# Dimensional reduction and incommensurate dynamic correlations in the $S = \frac{1}{2}$ triangular-lattice antiferromagnet Ca$_3$ReO$_5$Cl$_2$

S. A. Zvyagin [1] ✉, A. N. Ponomaryov[1,4], J. Wosnitza[1,2], D. Hirai [3], Z. Hiroi [3], M. Gen[3], Y. Kohama [3], A. Matsuo[3], Y. H. Matsuda [3] & K. Kindo[3]

The observation of spinon excitations in the $S = \frac{1}{2}$ triangular antiferromagnet Ca$_3$ReO$_5$Cl$_2$ reveals a quasi-one-dimensional (1D) nature of magnetic correlations, in spite of the nominally 2D magnetic structure. This phenomenon is known as frustration-induced dimensional reduction. Here, we present high-field electron spin resonance spectroscopy and magnetization studies of Ca$_3$ReO$_5$Cl$_2$, allowing us not only to refine spin-Hamiltonian parameters, but also to investigate peculiarities of its low-energy spin dynamics. We argue that the presence of the uniform Dzyaloshinskii-Moriya interaction (DMI) shifts the spinon continuum in momentum space and, as a result, opens a zero-field gap at the Γ point. We observed this gap directly. The shift is found to be consistent with the structural modulation in the ordered state, suggesting this material as a perfect model triangular-lattice system, where a pure DMI-spiral ground state can be realized.

$S = \frac{1}{2}$ antiferromagnetic systems with triangular structures are in the focus of modern quantum physics, in particular, in connection with Anderson's idea of "resonating valence bond" states in frustrated spin systems[1]. He proposed that the corresponding ground state can be a two-dimensional (2D) fluid of resonating spin-singlet pairs, with the elementary excitation spectrum formed by fractionalized mobile quasiparticles, spinons. Such excitations were observed in the spatially anisotropic triangular-lattice antiferromagnet (AF) Cs$_2$CuCl$_4$[2,3], suggesting that the spin-liquid scenario is indeed realized in this material. On the other hand, more recent analysis of the inelastic neutron scattering data[4] unveiled a quasi-1D nature of magnetic correlations in Cs$_2$CuCl$_4$, in spite of its nominally 2D magnetic structure. This phenomenon is known as frustration-induced dimensional reduction[5]. Later, electron spin resonance (ESR) spectroscopy studies of Cs$_2$CuCl$_4$ in the magnetically disordered state revealed the presence of an energy gap[6] at the Γ point, corresponding to a shift of the spinon continuum in momentum space, as predicted for an $S = \frac{1}{2}$ isotropic Heisenberg AF chain with

uniform antisymmetric exchange interaction (also known as uniform Dzyaloshinskii-Moriya interaction; DMI)[7]. Recently, the quasi-1D nature of spin correlations in Cs$_2$CuCl$_4$ has been independently confirmed by thermal-transport measurements[8]. It was shown that the uniform DMI remains to play an important role also below $T_N$, favoring a noncollinear helical spin structure[9].

Such DMI-induced incommensurate magnetic structures have recently attracted great attention, hosting a number of intriguing phenomena (e.g., magnetoelectric effects in multiferroics[10,11] and magnetic skyrmions[12]), which have direct relevance to various potential technological applications, including sensors, magnetic-memory devices, etc. Since competing nearest- and next-nearest-neighbor exchange interactions can be another source of magnetic incommensurability[13,14], it is important to distinguish between these two principally different mechanisms, which can coexist in real materials[15]. Searching for new model materials with non-collinear incommensurate structures, where spin correlations are determined solely by DMI (i.e., without or with a minimal admixture of the

[1]Dresden High Magnetic Field Laboratory (HLD-EMFL) and Würzburg-Dresden Cluster of Excellence ct.qmat, Helmholtz-Zentrum Dresden-Rossendorf, 01328 Dresden, Germany. [2]Institut für Festkörper- und Materialphysik, TU Dresden, 01062 Dresden, Germany. [3]Institute for Solid State Physics, University of Tokyo, Kashiwa, Chiba 277-8581, Japan. [4]Present address: Institute of Radiation Physics, Helmholtz-Zentrum Dresden-Rossendorf, 01328 Dresden, Germany. ✉e-mail: s.zvyagin@hzdr.de

competing exchange interactions) remains a very important task, both from the scientific as well as application perspective.

The recently synthesized compound $Ca_3ReO_5Cl_2$ (CROC hereafter) is known for its unusually pronounced pleochroism[16], exhibiting different colors depending on the viewing direction. The compound crystallizes in an orthorhombic *Pmna* ($Z = 4$) structure with magnetic $Re^{6+}$ ions arranged in triangular-lattice structures in the *bc* plane (Fig. 1)[17,18]. Each $Re^{6+}$ ion is surrounded by five oxide ions, forming a $ReO_5$ square-pyramidal unit (Fig. 1a). The crystal field from the neighboring oxide and chloride ions lifts the degeneracy of the Re $5d^1$ levels, stabilizing the effective $S = \frac{1}{2}$ $d_{xy}$ orbital ground state with the $d_{xy}$ orbitals confined in the *bc* plane. The orbitals overlap with each other, resulting in a large direct exchange interaction $J$ along the *b* axis (this axis is the chain direction). A small orbital overlap between adjacent chains results in the weaker interchain exchange interaction $J'$. The $Ca_3ReO_5$ layers are well separated from each other by Cl layers. First-principle density functional theory (DFT) calculations revealed that the interplane exchange interaction $J''$ is more than three orders of magnitude smaller than the leading interaction $J$[17], allowing to map CROC to a quasi-2D triangular-lattice AF model. $ReO_5$ pyramidal units from the same chain point in the same direction, while units from adjacent chains point in opposite direction. As a result, there is no inversion center between neighboring Re atoms along the chains, allowing uniform DMI on the *b*-bond.

Recent inelastic neutron-scattering studies of this material revealed the presence of a two-spinon continuum[19], confirming the dimensional reduction in this $S = \frac{1}{2}$ AF with triangular-lattice structure. Similar to $Cs_2CuCl_4$, the quasi-1D character of the magnetic correlations is evident from a particular pronounced dispersion of magnetic excitations along the *b* axis and a distinct asymmetry of the neutron-scattering intensities in momentum space (a signature of bound spinon excitations, triplons). The quasi-1D nature of magnetic excitations in CROC was confirmed by means of Raman scattering[20]. Neutron-diffraction measurements revealed an incommensurate magnetic structure in CROC below $T_N = 1.13$ K, with the ordering wavevector $\mathbf{q} = (0, 0.465, 0)$[19].

Compared to $Cs_2CuCl_4$ with $J/k_B = 4.7$ K and $J'/J \simeq 0.30$[21,22], CROC is characterized by about one order of magnitude larger scale of exchange interactions. Fit of the magnetic susceptibility using an anisotropic triangular-lattice model unveiled $J/k_B = 40.6$ K and $J'/J = 0.32$, while a Bonner-Fisher fit for the $S = \frac{1}{2}$ Heisenberg AF chain model, combined with results of DFT calculations, suggested $J/k_B = 41.3$ K and $J'/J = 0.295$[17]. On the other hand, recent inelastic neutron-scattering measurements revealed $J/k_B = 41.7$ K and $J'/J = 0.15(5)$[19]. Thus, although the intrachain exchange parameters are in good agreement with each other, an independent and, preferably, more accurate estimation of $J'$ in CROC remains a challenging open problem. Here, we present high-field ESR spectroscopy and magnetization studies of CROC, allowing us not only to refine its spin-Hamiltonian parameters, but also to investigate the magnetic properties and peculiarities of spin dynamics in a broad range of frequencies and magnetic fields, relevant to the energy scale of magnetic interactions in this new frustrated spin system.

## Results

### High-field ESR

In Fig. 2, we show the frequency-field diagrams of the ESR excitations measured at a temperature of 2 K with magnetic field applied along the *a*, *b*, and *c* axes. We estimate the accuracy of sample orientation to be better than ±5° (see Supplementary Information). We observed two ESR modes for fields applied along the *a* and *b* axes (A1, A2, and B1, B2, respectively), while we detected three modes (C1, C2, C2') for the field along the *c* axis. The extrapolation of the frequency-field dependences to zero field reveals a gap of $\Delta = 310(5)$ GHz. In Fig. 3, we show examples of low-temperature ESR spectra. We measured also the temperature dependences of the resonance fields for the modes A1 and A2 at a frequency of 396 GHz (Fig. 4).

### High-field magnetization

In Fig. 5, we show the magnetization of a CROC powder sample in magnetic fields up to 120 T, obtained using a pulsed single-turn magnet (red solid line). The left inset shows the derivative of the as-measured high-field magnetization, revealing clearly the saturation field of $\mu_0 H_{sat} = 83.6$ T. In addition, we performed magnetization of the

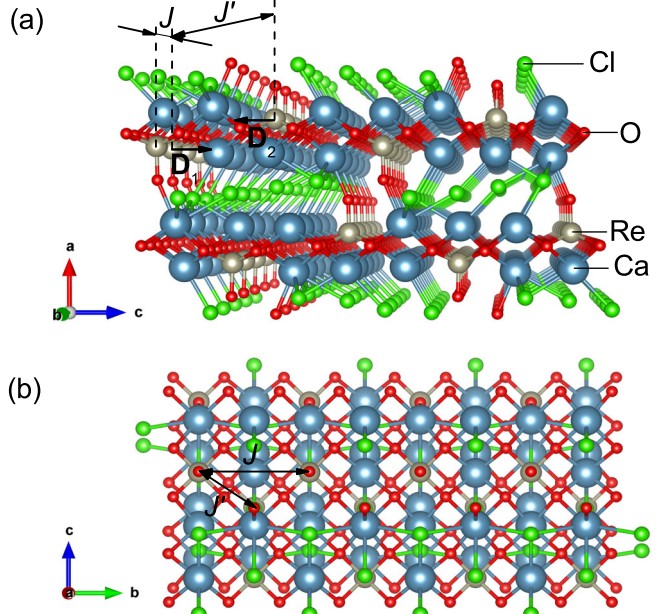

**Fig. 1 | Crystal structure of CROC. a** Perspective view along the $Re^{6+}$ chain direction (*b* axis). Ca atoms are shown in blue, Re in gray, O in red, and Cl in green. $J$ and $J'$ are intra- and interchain exchange interactions, respectively. The DMI vectors from adjacent chains ($\mathbf{D}_1$ and $\mathbf{D}_2$) are schematically shown by arrows (see the text for details). **b** Parallel view of the crystal structure of CROC along the *a* axis.

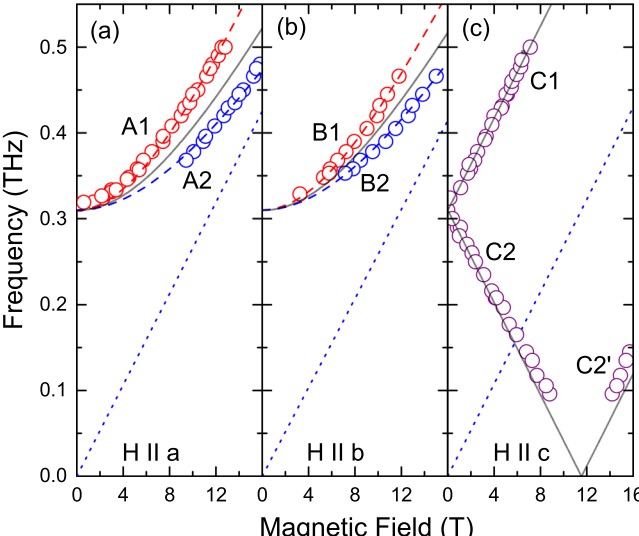

**Fig. 2 | Frequency-field diagrams of magnetic excitations in CROC.** Fields are applied along the *a* (**a**), *b* (**b**), and *c* (**c**) axis (*T* = 2 K). The solid lines are results of calculations using Eqs. (3) and (4). The dotted lines correspond to the paramagnetic *g*-factors, 1.88, 1.85, and 1.92 for field applied along the *a*, *b*, and *c* axis, respectively[19]. The dashed lines are guides for the eye.

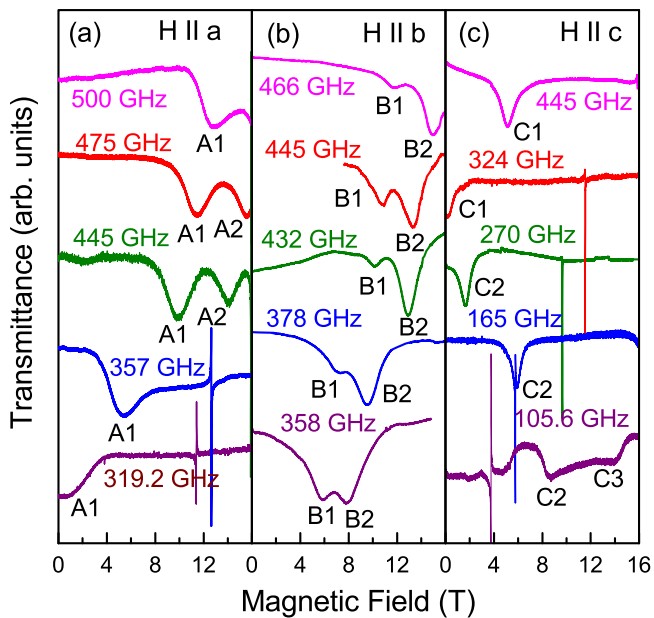

**Fig. 3 | Exemplary ESR spectra.** Fields are applied along the $a$ (**a**), $b$ (**b**), and $c$ (**c**) axis ($T = 2$ K). Narrow lines with $g = 2$ correspond to DPPH, used as a marker.

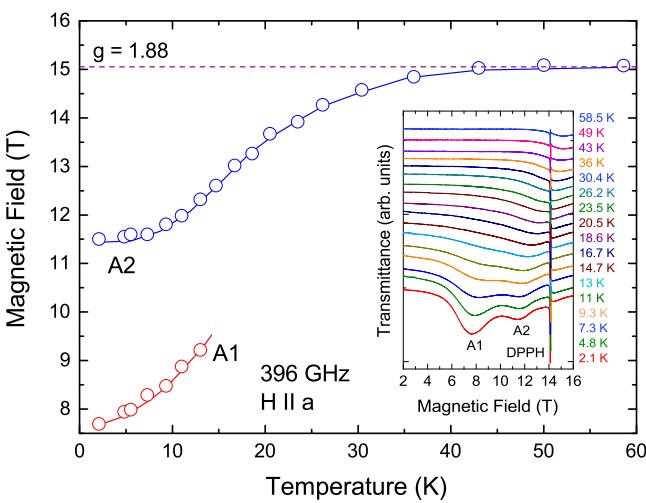

**Fig. 4 | Temperature dependences of ESR fields in CROC.** The data are taken at a frequency of 396 GHz with magnetic field applied along the $a$ axis. The dashed line corresponds to the paramagnetic $g = 1.88$. The solids lines are guides for the eye. The inset shows corresponding spectra at different temperatures (narrow lines with $g = 2$ correspond to DPPH, used as a marker).

measurements of a CROC powder sample in magnetic fields up to 51 T using a nondestructive magnet (blue line, right inset in Fig. 5). The employment of this magnet allowed us to achieve much better signal-to-noise ratio and to identify low-field features of the 120 T magnetization as artifacts. In addition, we measured the magnetization of a powder sample at a temperature of 2 K in DC fields up to 7 T (red line). The experimental data were compared with the results of calculations obtained using the orthogonalized finite-temperature Lanczos method (OFTLM) for a triangular-lattice AF[23].

## Discussion

The spin dynamics in an $S = \frac{1}{2}$ Heisenberg AF chain with uniform nearest-neighbor exchange coupling $J$ is determined by a gapless two-particle continuum of fractional $S = \frac{1}{2}$ excitations, spinons. The energy of the lower boundaries of the continuum is given by the des

Cloizeaux-Pearson relation[24,25]

$$\varepsilon_l(q) = \frac{\pi}{2}J|\sin(q)|, \tag{1}$$

and the upper boundaries are described by the formula[26]

$$\varepsilon_u(q) = \pi J|\sin(q/2)|, \tag{2}$$

where $q$ is the wavevector along the chain direction.

The presence of uniform DMI in an $S = \frac{1}{2}$ Heisenberg AF chain can significantly modify the excitation spectrum, shifting the spinon continuum in momentum space[7]. As result, an energy gap at the Brillouin-zone center opens. Such a gap was observed by means of ESR in the "dimensionally reduced" triangular-lattice AF $Cs_2CuCl_4$[6], quasi-1D Heisenberg AFs $Na_2CuSO_4Cl_2$[27], $K_2CuSO_4Br_2$[28], and $K_2CuSO_4Br_2$[29]. The employment of ESR techniques allows one not only to directly measure the uniform DMI, but also to experimentally investigate the interaction between fractionalized spinon excitations, including backscattering processes[30,31].

The theory of ESR in an $S = \frac{1}{2}$ Heisenberg AF chain with uniform DMI[6,28,31] predicts the presence of two or one ESR modes, depending on field direction. The excitation frequency-field diagram for $\mathbf{H} \| \mathbf{D}$ is given by

$$h\nu_\pm = |g_\| \mu_B H \pm \frac{\pi D}{2}| \tag{3}$$

(thereby, $D$ is a parameter describing the DMI strength), while for $\mathbf{H} \perp \mathbf{D}$

$$h\nu = \sqrt{(g_\perp \mu_B H)^2 + \left(\frac{\pi D}{2}\right)^2}. \tag{4}$$

As mentioned above, due to the absence of an inversion center between adjacent $Re^{6+}$ ions along the chains, DMI in CROC is allowed along the $b$ direction (Fig. 1). There is a mirror plane perpendicular to the chains with a bisecting point located in the middle of the section connecting two neighboring $Re^{6+}$ ions. Therefore, in accordance with Moriya's rules[32], the DM vector is in the $ac$ plane (Fig. 1). It is important to mention that the $ReO_5$ pyramidal units are not connected with each other by sharing common oxygen atoms. Instead, $Re^{6+}$ ions along the chains are linked by a more complex superexchange path, formed by four basal oxygen and two calcium atoms. The path has a mirror plane, which includes one oxygen ion is in the pyramid apex and two neighboring rhenium ions. Because of that, the DM vector is expected to be perpendicular to the mirror plane[32], i.e., $\mathbf{D} \| \mathbf{c}$. For $\mathbf{H} \| \mathbf{D}$, the theory[6,28,31] predicts ESR modes as described by Eq. (3). These modes were indeed observed in our experiments with $\mathbf{H} \| \mathbf{c}$ (Fig. 2c).

For the relevant magnetic-field range with the field applied perpendicular to $\mathbf{D}$, the theory predicts the presence of one ESR mode (Fig. 2a, b, solid lines). On the other hand, our analysis of the ESR angular dependence revealed that even a small (a few degree) field tilting from the $b$ towards the $c$ direction may result in an ESR line splitting, as observed by us for all frequencies above 350 GHz (as an example, ESR angular dependences at 370 GHz with magnetic field in the $bc$ plane are presented in Supplementary Information). Most importantly, the extrapolations of the ESR frequency-field dependences to zero field suggest the presence of a zero-field gap, $\Delta = 310$ GHz. Employing Eqs. (3) and (4) we can estimate DMI, yielding $D/k_B = 9.47$ K. The temperature dependences of the resonance fields for modes A1 and A2 (Fig. 4) revealed that with increasing temperature these modes come so close to each other, that above 13 K one can resolve only one line. On the other hand, the ESR absorption shifts to higher fields, reaching the paramagnetic value $g = 1.88$ at about 40 K (dashed

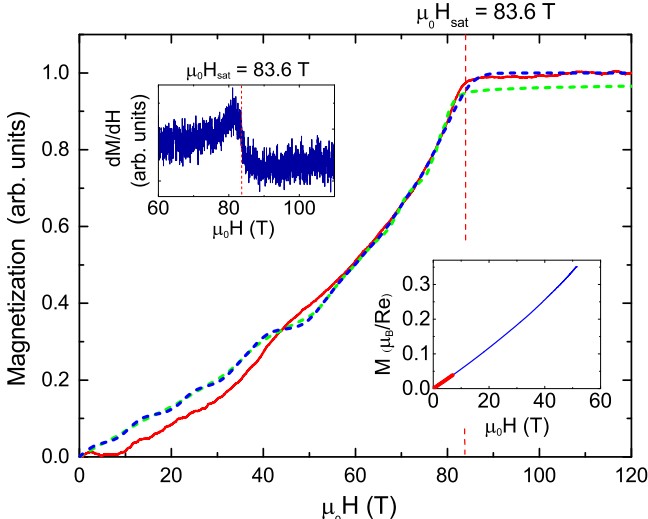

**Fig. 5 | Magnetization of a CROC.** Magnetization of a CROC powder sample in magnetic fields up to 120 T, obtained using a pulsed single-turn magnet (red line; the initial temperature is 5 K) together with results of OFTLM calculations for a triangular-lattice AF with $J/k_B = 41.7$ K, $J'/J = 0.25$ in isothermal (blue dashed line) and adiabatic (green dashed line) approximations[23]. The left inset shows the derivative of the as-measured magnetization $M$; the saturation field $\mu_0 H_{sat} = 83.6$ T is determined as shown. The right inset shows the magnetization of a CROC powder sample in magnetic fields up to 51 T measured using a non-destructive magnet (the initial temperature is 1.3 K). Magnetization of a powder sample at a temperature of 2 K in DC fields up to 7 T is shown in red.

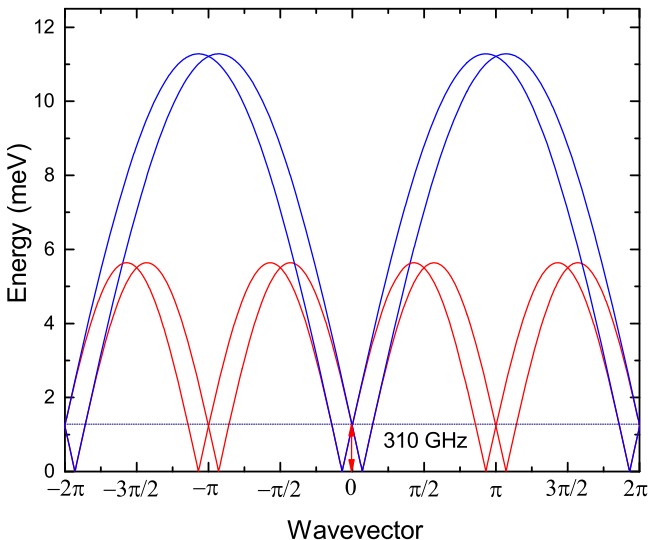

**Fig. 6 | Two-spinon continuum calculated for an $S = \frac{1}{2}$ Heisenberg AF chain with $J/k_B = 41.7$ K and the uniform DMI $D/k_B = 9.47$ K.** The upper and lower boundaries of the two-spinon continuum are denoted by the blue and red lines, respectively. The gap observed at the $\Gamma$ point, $\Delta = 310$ GHz, is shown by the arrow.

line in Fig. 4). This temperature corresponds to the energy of the intrachain exchange interaction $J$; above this temperature thermal fluctuations become dominant, significantly suppressing spin correlations along the chain direction. Investigating temperature and angular dependences of ESR parameters (which in CROC are dominantly determined by the exchange interaction $J$ and uniform DMI $D$) remains a topic of further experimental and theoretical investigations.

Knowing the saturation field and intrachain exchange coupling, we now can determine the interchain exchange interaction $J'$ from the

expression

$$g\mu_B H_{sat} = 2J(1 + J'/2J)^2, \qquad (5)$$

obtained from the exchange model of a triangular-lattice AF[22]. Using the averaged value $\langle g \rangle = 1.883$ for the powder sample, $J/k_B = 41.7$ K from inelastic neutron scattering[19], and $\mu H_{sat} = 83.6$ T (Fig. 5), we obtain $J'/k_B = 10.5$ K and $J'/J = 0.25$.

In Fig. 5, together with experimental magnetization data (red line) we show also OFTLM calculation results for a triangular-lattice AF with $J/k_B = 41.7$ K and $J'/J = 0.25$[23]. The data for an isothermal ($k_B T/J = 0.05$; $N = 36$ sites) approximation are shown by the blue dashed line. It is important to mention that, due to the relatively small pulse duration, magnetization processes in pulsed fields in the megagauss range are not isothermal, but rather adiabatic. The adiabatic magnetization for $J/k_B = 41.7$ K, $J'/J = 0.25$, and $S_m/N = 0.075$ (where $S_m$ is the magnetic anisotropy and $N = 36$ sites) is shown in Fig. 5 by the green dashed line. Very good agreement between the experimental data and the calculation is achieved. For the adiabatic process at a given initial temperature (-0.1 J/$k_B$) the theory suggests that the ground state at $H_{sat}$ is only partially spin-polarized (green dash line in Fig. 5.

In Fig. 6, we schematically show a two-spinon continuum for $S = \frac{1}{2}$ Heisenberg AF chain system with $J/k_B = 41.7$ K and the $\Gamma$-point gap $\Delta = 310$ GHz, as revealed by ESR. For these values, the shift of the soft mode in the magnetically disordered phase (determined by the uniform DMI, as discussed above) corresponds to the incommensurate wavevector $q = 0.464$, which is remarkably consistent with the ordering wavevector $q = (0, 0.465, 0)$ below $T_N$[19]. This should be regarded as an important prerequisite for the realization of a pure DMI-spiral magnetically ordered state in this material at low temperatures.

The situation is very different from the case of Cs$_2$CuCl$_4$, where the incommensurate ordered structure appears to be a combined effect of the frustration, induced by exchange interaction between the spins along chains[9], and DMI. Assuming $J/k_B = 4.7$ K[21,22] and $\Delta = 14$ GHz[6], we obtain for the soft mode in the magnetically disordered phase $q = 0.485$. This value is somewhat different from the $b$ component of the ordering wavevector $q = (0, 0.472, 0)$[9], which results in a twice larger gap at the $\Gamma$ point[33]. The difference between Cs$_2$CuCl$_4$ and CROC can be understood, taking into account that CROC is characterized by an about five times larger frustration factor $f = |\Theta_{cw}|/T_N$ (33.5 vs 6.5 for CROC and Cs$_2$CuCl$_4$, respectively[17,34]), where $\Theta_{cw}$ is the Curie-Weiss temperature. The larger frustration in CROC leads to more effective isolation of neighboring chains from each other, with the uniform DMI playing the key role above and below $T_N$.

In summary, we performed high-field ESR spectroscopy and magnetization studies of CROC allowing us to characterize this material as a spatially anisotropic triangular-lattice Heisenberg AF with $J'/J = 0.25$, frustration-induced dimensional reduction, and the incommensurate spin dynamics. Our findings illuminate the important role of the uniform DMI in this material, affecting the spin dynamics in the magnetically disordered state and determining peculiarities of its magnetic structure in the magnetically ordered phase. Our observations suggest, that a pure DMI-spiral state can be realized in CROC, making this material an attractive toy model to explore details of the dimensional reduction and other effects of the geometrical frustration in low-D spin systems with competing interactions.

## Methods
### Single-crystal growth
Single crystals of CROC were grown by a flux method[18]. In an argon-filled glovebox, CaO, ReO$_3$, and CaCl$_2$ were mixed in an agate mortar in a 4.1:1:17 molar ratio, then placed in a gold tube and sealed in an evacuated quartz ampule. The ampule was heated to 1000 °C and kept

for 24 h before being cooled down to 800 °C at a rate of 1 °C per hour. Single crystals were obtained after washing away excess $CaCl_2$ flux with distilled water. Single crystals with typical sizes of ca $4 \times 4 \times 1$ mm³ were used in ESR experiments. The crystal's cleavage plane is perpendicular to the $a$ axis. It is important to mention that the single crystals gradually decompose in air due to moisture.

### High-field ESR
High-field ESR measurements of CROC were performed at the Dresden High Magnetic Field Laboratory (HLD) using a transmission-type ESR spectrometer (similar to that described in ref. 35) with oversized waveguides and a 16 T superconducting magnet. A set of backward-wave oscillators, Gunn diodes, and VDI microwave sources (Virginia Diodes Inc, USA) was used, allowing us to study magnetic excitations in a broad quasicontinuously covered frequency range, from 100 to 500 GHz. The experiments were performed in the Faraday and Voigt configurations with magnetic fields $H\|a$ and $H\|b,c$, respectively. The stable free-radical 2,2-diphenyl-1-picrylhydrazyl (DPPH) was used as a frequency-field marker.

### High-field magnetization
The megagauss-field facility at the ISSP, University of Tokyo, Japan, equipped with a 160 $\mu$F/50 kV capacitor bank, was used to generate ultrahigh magnetic fields[36,37]. To measure magnetization, we employed two types of magnet. For 120 T experiments, we used single-turn coil magnets in horizontal geometry, with the pulse duration 8 $\mu$s. The induction method was used to detect the $dM/dH$ signal using coaxial-type pickup coils. The field was measured by a calibrated pickup coil located near the sample. The recorded pickup voltage was numerically integrated to obtain field values. In spite of careful compensation, the pulsed-field magnetization exhibited a tiny linear background, which we subtracted from the experimental data. Magnetization up to 51 T was measured using a non-destructive magnet with a pulse duration of 4 ms. More detailed information on the experimental procedure can be found in ref. 38. The magnetization was calibrated to absolute values using the magnetization data collected at 2 K in DC fields up to 7 T using a SQUID magnetometer (Quantum Design, USA).

## Data availability
The data that support the findings of this study are available from the corresponding author upon reasonable request.

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

## Acknowledgements

This work was supported by the Deutsche Forschungsgemeinschaft, through ZV 6/2-2, the Würzburg-Dresden Cluster of Excellence on Complexity and Topology in Quantum Matter - *ct.qmat* (EXC 2147, project No. 390858490), and SFB 1143, as well as by HLD at HZDR, member of the European Magnetic Field Laboratory (EMFL). S.Z. acknowledges the support of the BMBF via DAAD (Project ID. 57457940). This work was partly supported by the Japan Society for the Promotion of Science (JSPS) KAKENHI Grant Numbers JP19H04688 and JP20H01858. S.Z. would like to acknowledge fruitful discussions with A. I. Smirnov, K. Yu. Povarov, and K. Nawa. We would like to thank K. Morita, who shared with us results of his numerical calculations.

## Author contributions

S.Z. conceived, designed, and led the project. D.H. and Z.H. grew CROC single crystals. S.Z. and A.P. performed high-field ESR experiments. M.G., Y.K., A.M., Y.M., and K.K. performed high-field magnetization experiments. J.W. administered the HLD parts of the project. All authors discussed the results and commented on the manuscript.

## Funding

## Competing interests

The authors declare no competing interests.
