## [Peer Review File · Nature Communications]

REVIEWER COMMENTS

Reviewer #1 (Remarks to the Author):

In their manuscript Zvyagin et al. report the observation of the unusual ESR response of a novel quantum material CROC. This material represents a $S=1/2$ antiferromagnet on a distorted triangular lattice, with unequal exchanges J along the base, and J' along the sides. The exchange J is dominant, allowing to think of the system as a collection of relatively weakly coupled chains.

The empirically observed splittings (at temperature much lower than the exchange $J \sim 40$ K) point at the effect of uniform Dzyaloshinskii-Moriya interaction on the chain bonds. Specifically, they observe one (correspondingly, two) ESR lines when the magnetic field is applied perpendicular (correspondingly, parallel) to the c -axis of the material. This pattern of resonance lines has been earlier predicted theoretically and observed in model materials Cs_2CuCl_4 and $\text{K}_2\text{CuSO}_4\text{Br}_2$, as well as in $\text{K}_2\text{CuSO}_4\text{Cl}_2$ and, most recently, $\text{Na}_2\text{CuSO}_4\text{Cl}_2$.

These data, and in particular the frequency of the zero-field resonance (310 GHz) and the field at which the downward-going component of the ESR doublet (for the $H \parallel c$ -axis geometry, Fig.1c), allow authors to directly deduce the magnitude of in-chain DM interaction. They also observe that in the transverse geometry (field along the a - and b -axis) the ESR mode is gapped (as predicted by the theory, Eq.(3)) but in addition shows a small splitting into A_1/A_2 and B_1/B_2 modes as shown in Fig.1(a,b). In addition, the pulsed-field magnetization of CROC is measured up to 120 T, showing the saturation field of ~ 83 T, which is used to estimate the frustrated interchain exchange J' in Eq.(5).

The study is clearly a step forward in understanding this interesting and exotic material, and as such has a potential to be publishable by Nature Communications. However, we have several concerns regarding the data and the analysis that should be addressed before proceeding further.

1) Can some symmetry analysis be carried out in order to predict which DM orientations are actually symmetry-allowed? Is the DM vector being precisely along c -axis, as stated in the manuscript, an accident or is it a consequence of the crystal structure/symmetry? Fig. 1 is not really helpful for figuring this out because it does not even show the main J bond. A clearer figure showing the geometry of exchange interactions and directions of the allowed DM vectors is very much needed.

2) While the sample misalignment can indeed be the case for extra line splitting (A_1/A_2 and B_1/B_2) in the transverse orientation, the identical pattern of this splitting for two independent orientations

($H \parallel a$ and $H \parallel b$) is surprising. Since the authors mention that the angle-dependent ESR data has been measured, see Ref.[28], we'd like to ask to show this data in the paper, or in the supplemental material.

3) Since the sample alignment is an issue, see Ref.[23], it would be helpful to mention typical sample dimensions and shapes.

4) We would like to point out two recent papers that are directly relevant to the discussed physics of the quantum spin chain with the uniform DM interaction. The first is the experimental study of $\text{Na}_2\text{CuSO}_4\text{Cl}_2$ by Fujihala et al., Phys.Rev. B 105 144410 (2022). And the second is an even more recent experimental investigation of the ESR in the $H \parallel$ DM geometry in PRL 128, 187202 (2022), which led to the determination of the magnitude of the inter-spinon interaction in $\text{K}_2\text{CuSO}_4\text{Br}_2$.

5) We'd like to point out comparable magnitudes of the DM (9.5 K) and interchain interaction J' (9.3 K), as determined by the authors, make the analysis of the propagation wave vector of the incommensurate spiral ($q=0.464$) potentially less straightforward. That is, the magnitude of q may depend on both J' and D in some non-trivial manner.

6a) The initial segment of the magnetization curve in Fig.5, up to about 10 T, appears to be very flat. Is this suppression real or is it just a pulsed-field measurement artifact? The intriguing point here is that the field at which the magnetization appears to take off seems to coincide with the softening field of the DM-split mode in Fig.1c.

6b) The magnetization curve also shows a 'bump' around 1/3 of the full magnetization. Is this effect real, or is it also a measurement artifact?

Minor points:

- There appears to be an error with references above Eq.(2).

Reviewer #2 (Remarks to the Author):

Remarks to the Author:

In their manuscript Zvyagin et al. report the observation of the unusual ESR response of a novel quantum material CROC. This material represents a $S=1/2$ antiferromagnet on a distorted triangular lattice, with unequal exchanges J along the base, and J' along the sides. The exchange J is dominant, allowing to think of the system as a collection of relatively weakly coupled chains.

The empirically observed splittings (at temperature much lower than the exchange $J \sim 40$ K) point at the effect of uniform Dzyaloshinskii-Moriya interaction on the chain bonds. Specifically, they observe one (correspondingly, two) ESR lines when the magnetic field is applied perpendicular (correspondingly, parallel) to the c -axis of the material. This pattern of resonance lines has been earlier predicted theoretically and observed in model materials Cs_2CuCl_4 and $\text{K}_2\text{CuSO}_4\text{Br}_2$, as well as in $\text{K}_2\text{CuSO}_4\text{Cl}_2$ and, most recently, $\text{Na}_2\text{CuSO}_4\text{Cl}_2$.

These data, and in particular the frequency of the zero-field resonance (310 GHz) and the field at which the downward-going component of the ESR doublet (for the $H \parallel c$ -axis geometry, Fig.1c), allow authors to directly deduce the magnitude of in-chain DM interaction. They also observe that in the transverse geometry (field along the a - and b -axis) the ESR mode is gapped (as predicted by the theory, Eq.(3)) but in addition shows a small splitting into A_1/A_2 and B_1/B_2 modes as shown in Fig.1(a,b). In addition, the pulsed-field magnetization of CROC is measured up to 120 T, showing the saturation field of ~ 83 T, which is used to estimate the frustrated interchain exchange J' in Eq.(5).

The study is clearly a step forward in understanding this interesting and exotic material, and as such has a potential to be publishable by Nature Communications. However, we have several concerns regarding the data and the analysis that should be addressed before proceeding further.

1) Can some symmetry analysis be carried out in order to predict which DM orientations are actually symmetry-allowed? Is the DM vector being precisely along c -axis, as stated in the manuscript, an accident or is it a consequence of the crystal structure/symmetry? Fig. 1 is not really helpful for figuring this out because it does not even show the main J bond. A clearer figure showing the geometry of exchange interactions and directions of the allowed DM vectors is very much needed.

2) While the sample misalignment can indeed be the case for extra line splitting (A_1/A_2 and B_1/B_2) in the transverse orientation, the identical pattern of this splitting for two independent orientations ($H \parallel a$ and $H \parallel b$) is surprising. Since the authors mention that the angle-dependent ESR data has been measured, see Ref.[28], we'd like to ask to show this data in the paper, or in the supplemental material.

3) Since the sample alignment is an issue, see Ref.[23], it would be helpful to mention typical sample dimensions and shapes.

4) We would like to point out two recent papers that are directly relevant to the discussed physics of the quantum spin chain with the uniform DM interaction. The first is the experimental study of $\text{Na}_2\text{CuSO}_4\text{Cl}_2$ by Fujihala et al., Phys.Rev. B 105 144410 (2022). And the second is an even more recent experimental investigation of the ESR in the $H \parallel DM$ geometry in PRL 128, 187202 (2022), which led to the determination of the magnitude of the inter-spinon interaction in $\text{K}_2\text{CuSO}_4\text{Br}_2$.

5) We'd like to point out comparable magnitudes of the DM (9.5 K) and interchain interaction J' (9.3 K), as determined by the authors, make the analysis of the propagation wave vector of the incommensurate spiral ($q=0.464$) potentially less straightforward. That is, the magnitude of q may depend on both J' and D in some non-trivial manner.

6a) The initial segment of the magnetization curve in Fig.5, up to about 10 T, appears to be very flat. Is this suppression real or is it just a pulsed-field measurement artifact? The intriguing point here is that the field at which the magnetization appears to take off seems to coincide with the softening field of the DM-split mode in Fig.1c.

6b) The magnetization curve also shows a 'bump' around 1/3 of the full magnetization. Is this effect real, or is it also a measurement artifact?

Minor points:

- There appears to be an error with references above Eq.(2).

Reviewer #3 (Remarks to the Author):

S. A. Zvyagin and coworkers report the high-field ESR and magnetization investigations of the anisotropic triangular lattice AFM $\text{Ca}_3\text{ReO}_5\text{Cl}_2$. The studied compound constitutes the second example that exhibits a frustration-induced dimensional reduction after Cs_2CuCl_4 . Previous studies

established a quasi-1D nature of magnetism and an incommensurate magnetic ordering at $T_N=1.13$ K. The key findings are the presence of the uniform DM interaction, engendering a zero-field gap at the Gamma point, and a spiral ground state. In addition, the identification of the saturation field enables the authors to refine the magnetic parameters.

I judge that the manuscript is well written and the discovery of an only DMI-induced spiral ground state is interesting. But the experimental data and their analysis are not convincing enough for publication of the manuscript in Nature Communications. In addition to the influence of DMI on low-energy excitations, the evolution of spin correlations through dimensional reduction should be addressed.

In addition, the manuscript should be amended in some passages where the description of the data is either unclear or missing.

(1) ESR: The authors concentrated on the frequency dependence of ESR. For a complete presentation of the ESR data, the temperature, angle, and orientation dependences should be discussed. For example, the temperature dependence of the ESR linewidth between the intra and interchain directions may contain valuable information about distinct spin dynamics through dimensional reduction. However, I find that the presented ESR data are marginal for a high-impact journal.

(2) Magnetization: The numerical simulation based on the determined magnetic parameters should be compared with the experimental data.

10.08.2022

Dear Reviewers,

Thanks a lot for the time spent with our manuscript. Please find attached the revised version of the manuscript. As you can see, the manuscript has been substantially modified, in accordance with your comments and suggestions. The concerns were systematically addressed with additional work, including lower-noise magnetization and angular-dependent ESR measurements, more detailed symmetry analysis of the crystal structure of CROC, and the comparison of our experimental data with results of numerical calculations.

All the relevant changes are highlighted in red.

We hope that now, after the revision, our work will be accepted for the publication.

Please do not hesitate to contact me in case of any questions or further inquiries.

Let us now go through the comments.

With best regards,

Sergei Zvyagin.

REVIEWER COMMENTS

Reviewer #1 (Remarks to the Author):

In their manuscript Zvyagin et al. report the observation of the unusual ESR response of a novel quantum material CROC. This material represents a $S=1/2$ antiferromagnet on a distorted triangular lattice, with unequal exchanges J along the base, and J' along the sides. The exchange J is dominant, allowing to think of the system as a collection of relatively weakly coupled chains.

The empirically observed splittings (at temperature much lower than the exchange $J \sim 40$ K) point at the effect of uniform Dzyaloshinskii-Moriya interaction on the chain bonds. Specifically, they observe one (correspondingly, two) ESR lines when the magnetic field is applied perpendicular (correspondingly, parallel) to the c -axis of the material. This pattern of resonance lines has been earlier predicted theoretically and observed in model materials Cs_2CuCl_4 and $\text{K}_2\text{CuSO}_4\text{Br}_2$, as well as in $\text{K}_2\text{CuSO}_4\text{Cl}_2$ and, most recently, $\text{Na}_2\text{CuSO}_4\text{Cl}_2$.

These data, and in particular the frequency of the zero-field resonance (310 GHz) and the field at which the downward-going component of the ESR doublet (for the $H \parallel c$ -axis geometry, Fig.1c), allow authors to directly deduce the magnitude of in-chain DM interaction. They also observe that in the transverse geometry (field along the a - and b -axis) the ESR mode is gapped (as predicted by the theory, Eq.(3)) but in addition shows a small splitting into A_1/A_2 and B_1/B_2 modes as shown in Fig.1(a,b). In addition, the

pulsed-field magnetization of CROC is measured up to 120 T, showing the saturation field of ~ 83 T, which is used to estimate the frustrated interchain exchange J' in Eq.(5).

The study is clearly a step forward in understanding this interesting and exotic material, and as such has a potential to be publishable by Nature Communications. However, we have several concerns regarding the data and the analysis that should be addressed before proceeding further.

Our comment: We would like to thank Reviewer 1 and Reviewer 2 for the positive response. We agree with all the comments, trying to do our best when implementing the suggestions and remarks.

1) Can some symmetry analysis be carried out in order to predict which DM orientations are actually symmetry-allowed? Is the DM vector being precisely along c -axis, as stated in the manuscript, an accident or is it a consequence of the crystal structure/symmetry? Fig. 1 is not really helpful for figuring this out because it does not even show the main J bond. A clearer figure showing the geometry of exchange interactions and directions of the allowed DM vectors is very much needed.

Our comment: We have significantly modified Fig. 1. We show now two relevant projections, allowing one to better see the geometry of exchange interactions and the directions of the DM vectors, as follows from the symmetry analysis.

Due to the absence of inversion center between adjacent Re ions along chains, the uniform Dzyaloshinskii-Moriya interaction (DMI) in $\text{Ca}_3\text{ReO}_5\text{Cl}_2$ (CROC) is allowed along the b direction. To determine the DM vector, we employed Moriya's rules (Ref. 32). In accordance with the CROC structure:

(a) There is a mirror plane perpendicular to the chains running along the b axis with a bisecting point located in the middle of the section connecting two neighboring Re^{6+} ions. This makes evident that DM vectors are in the ac plane (having zero-component along the b axis).

(b) It is important to mention that two neighboring Re^{6+} ions along the chains are linked by a superexchange path, which also has a mirror plane (including the rhenium ions and one oxygen ion in the ReO_5 pyramid apex). In accordance with the Moriya's rules, the DM vector is expected to be perpendicular to this plane (i.e., $\mathbf{D} \parallel c$), which also perfectly agrees with our ESR data (based on the theory conclusions (Ref. 28, 29)).

2) While the sample misalignment can indeed be the case for extra line splitting (A1/A2 and B1/B2) in the transverse orientation, the identical pattern of this splitting for two independent orientations ($H \parallel a$ and $H \parallel b$) is surprising. Since the authors mention that the angle-dependent ESR data has been measured, see Ref.[28], we'd like to ask to show this data in the paper, or in the supplemental material.

Our comment: We have included in the manuscript the angular dependence of ESR fields in the bc plane (please see Supplementary Information in the end of the main text). Our symmetry analysis revealed that the corresponding DM vector does not have any component along the b axis, so that the observed ESR splitting (for $H \parallel b$, Fig. 2 (b)) is indeed due to a small misalignments of the applied magnetic field relative to the c direction. As mentioned in the manuscript, the accuracy of the sample orientation in these measurements was better than ± 5 degree, which includes the accuracy to the axis determinations, accuracy of the sample cut (unfortunately the sample does not have cleavage planes and needs to be cut), and the accuracy of the sample installation. This explains our observations.

Similar splitting was observed for $H \parallel a$, (Fig. 2 (a)), where, in accordance with the symmetry analysis, only one line is expected. Unfortunately, since the sample has a thin-plate-like shape (with the sample plate in the bc plane), studying ESR angular dependence only in the bc plane was possible.

3) Since the sample alignment is an issue, see Ref.[23], it would be helpful to mention typical sample dimensions and shapes.

Our comment: We have included this information in “Methods. Single-crystal growth” section.

4) We would like to point out two recent papers that are directly relevant to the discussed physics of the quantum spin chain with the uniform DM interaction. The first is the experimental study of $\text{Na}_2\text{CuSO}_4\text{Cl}_2$ by Fujihala et al., Phys.Rev. B 105 144410 (2022). And the second is an even more recent experimental investigation of the ESR in the $H \parallel$ DM geometry in PRL 128, 187202 (2022), which led to the determination of the magnitude of the inter-spinon interaction in $\text{K}_2\text{CuSO}_4\text{Br}_2$.

Our comment: Thank you. In accordance with the suggestion, we have added these references in the manuscript (Ref. 27, 30). We also added Ref. 29 (ESR in $\text{K}_2\text{CuSO}_4\text{Cl}_2$) and Ref. 31 (most recent and more detailed ESR theory for spin chains with uniform DMI).

5) We'd like to point out comparable magnitudes of the DM (9.5 K) and interchain interaction J' (9.3 K), as determined by the authors, make the analysis of the propagation wave vector of the incommensurate spiral ($q=0.464$) potentially less straightforward. That is, the magnitude of q may depend on both J' and D in some non-trivial manner.

Our comment: Surely, the magnitude of q can be affected by J' . The competition between these two parameters can be a very interesting topic of further theoretical and experimental research, once more materials with relevant properties are available. On the other hand, regarding the comment, we have to mention the following. The similar analysis, where the key role of DMI in the magnetically disordered phase was clearly established, was performed for another triangular-lattice antiferromagnet, Cs_2CuCl_4 (Ref. 6), with J'/D ratio about twice as large as in CROC. In addition, we argue that the influence of J' in CROC is much smaller due to the about five times larger frustration factor (if to compare to Cs_2CuCl_4). The larger frustration factor indeed leads to a much better chain isolation, reducing the potential effect of J' on q , and thus securing the validity of the employed approach.

6a) The initial segment of the magnetization curve in Fig.5, up to about 10 T, appears to be very flat. Is this suppression real or is it just a pulsed-field measurement artifact? The intriguing point here is that the field at which the magnetization appears to take off seems to coincide with the softening field of the DM-split mode in Fig.1c.

Our comment: As experimentalists, we have to admit here, that due to some technical issues all destructive magnets are, in general, much noisier than, e.g., non-destructive or DC magnets. Regarding the question: this is surely an artifact. To prove that, we have performed extra measurements, including the magnetization in magnetic fields up to 7 T (using DC-field SQUID magnetometer) and up to 51 T (employing a non-destructive magnet). The measurements have revealed the steady magnetization increase in the entire field range, without any flat region, as appeared in the destructive-coil data.

6b) The magnetization curve also shows a 'bump' around 1/3 of the full magnetization. Is this effect real, or is it also a measurement artifact?

Our comment: This is another artifact. The lower-noise magnetization measurement up to 51 T (employing a non-destructive magnet) does not reveal the presence of this "bump". On the other hand, it is important to mention that the theory (Ref. 23) predicts the presence of some anomaly corresponding to the 1/3 of saturation magnetization even for this relatively low J'/J ratio (reminiscent of the 1/3 magnetization plateau). We do plan to perform more accurate magnetization/ultrasound measurements of CROC using high-quality single crystals. The work is in progress.

Minor points:

- There appears to be an error with references above Eq.(2).

Our comments: We have fixed that. Thank you.

Reviewer #2 (Remarks to the Author):

Remarks to the Author:

In their manuscript Zvyagin et al. report the observation of the unusual ESR response of a novel quantum material CROC. This material represents a $S=1/2$ antiferromagnet on a distorted triangular lattice, with unequal exchanges J along the base, and J' along the sides. The exchange J is dominant, allowing to think of the system as a collection of relatively weakly coupled chains.

The empirically observed splittings (at temperature much lower than the exchange $J \sim 40$ K) point at the effect of uniform Dzyaloshinskii-Moriya interaction on the chain bonds. Specifically, they observe one (correspondingly, two) ESR lines when the magnetic field is applied perpendicular (correspondingly, parallel) to the c -axis of the material. This pattern of resonance lines has been earlier predicted theoretically and observed in model materials Cs_2CuCl_4 and $\text{K}_2\text{CuSO}_4\text{Br}_2$, as well as in $\text{K}_2\text{CuSO}_4\text{Cl}_2$ and, most recently, $\text{Na}_2\text{CuSO}_4\text{Cl}_2$.

These data, and in particular the frequency of the zero-field resonance (310 GHz) and the field at which the downward-going component of the ESR doublet (for the $H \parallel c$ -axis geometry, Fig.1c), allow authors to directly deduce the magnitude of in-chain DM interaction. They also observe that in the transverse geometry (field along the a - and b -axis) the ESR mode is gapped (as predicted by the theory, Eq.(3)) but in addition shows a small splitting into A_1/A_2 and B_1/B_2 modes as shown in Fig.1(a,b). In addition, the pulsed-field magnetization of CROC is measured up to 120 T, showing the saturation field of ~ 83 T, which is used to estimate the frustrated interchain exchange J' in Eq.(5).

The study is clearly a step forward in understanding this interesting and exotic material, and as such has a potential to be publishable by Nature Communications. However, we have several concerns regarding the data and the analysis that should be addressed before proceeding further.

1) Can some symmetry analysis be carried out in order to predict which DM orientations are actually symmetry-allowed? Is the DM vector being precisely along c -axis, as stated in the manuscript, an

accident or is it a consequence of the crystal structure/symmetry? Fig. 1 is not really helpful for figuring this out because it does not even show the main J bond. A clearer figure showing the geometry of exchange interactions and directions of the allowed DM vectors is very much needed.

2) While the sample misalignment can indeed be the case for extra line splitting (A1/A2 and B1/B2) in the transverse orientation, the identical pattern of this splitting for two independent orientations (H || a and H || b) is surprising. Since the authors mention that the angle-dependent ESR data has been measured, see Ref.[28], we'd like to ask to show this data in the paper, or in the supplemental material.

3) Since the sample alignment is an issue, see Ref.[23], it would be helpful to mention typical sample dimensions and shapes.

4) We would like to point out two recent papers that are directly relevant to the discussed physics of the quantum spin chain with the uniform DM interaction. The first is the experimental study of Na₂CuSO₄Cl₂ by Fujihala et al., Phys.Rev. B 105 144410 (2022). And the second is an even more recent experimental investigation of the ESR in the H || DM geometry in PRL 128, 187202 (2022), which led to the determination of the magnitude of the inter-spinon interaction in K₂CuSO₄Br₂.

5) We'd like to point out comparable magnitudes of the DM (9.5 K) and interchain interaction J' (9.3 K), as determined by the authors, make the analysis of the propagation wave vector of the incommensurate spiral (q=0.464) potentially less straightforward. That is, the magnitude of q may depend on both J' and D in some non-trivial manner.

6a) The initial segment of the magnetization curve in Fig.5, up to about 10 T, appears to be very flat. Is this suppression real or is it just a pulsed-field measurement artifact? The intriguing point here is that the field at which the magnetization appears to take off seems to coincide with the softening field of the DM-split mode in Fig.1c.

6b) The magnetization curve also shows a 'bump' around 1/3 of the full magnetization. Is this effect real, or is it also a measurement artifact?

Minor points:

- There appears to be an error with references above Eq.(2).

Reviewer #3 (Remarks to the Author):

S. A. Zvyagin and coworkers report the high-field ESR and magnetization investigations of the anisotropic triangular lattice AFM Ca₃ReO₅Cl₂. The studied compound constitutes the second example that exhibits a frustration-induced dimensional reduction after Cs₂CuCl₄. Previous studies established a quasi-1D nature of magnetism and an incommensurate magnetic ordering at TN=1.13 K. The key findings are the presence of the uniform DM interaction, engendering a zero-field gap at the Gamma point, and a spiral ground state. In addition, the identification of the saturation field enables the authors to refine the

magnetic parameters.

I judge that the manuscript is well written and the discovery of an only DMI-induced spiral ground state is interesting. But the experimental data and their analysis are not convincing enough for publication of the manuscript in Nature Communications. In addition to the influence of DMI on low-energy excitations, the evolution of spin correlations through dimensional reduction should be addressed.

Our comment: We would like to thank Reviewer 3 for the positive response. We agree with all the comments, trying to do our best when implementing the suggestions and remarks.

In addition, the manuscript should be amended in some passages where the description of the data is either unclear or missing.

(1) ESR: The authors concentrated on the frequency dependence of ESR. For a complete presentation of the ESR data, the temperature, angle, and orientation dependences should be discussed. For example, the temperature dependence of the ESR linewidth between the intra and interchain directions may contain valuable information about distinct spin dynamics through dimensional reduction. However, I find that the presented ESR data are marginal for a high-impact journal.

Our comment: We would like to point out that the term “dimensional reduction” in the context of our work (see also Ref. 4,5) does not mean the “temperature crossover”, but it is rather a structural peculiarity, discovered in Cs_2CuCl_4 , and more recently observed in CROC. Due to the geometrical frustration, the presence of quasi-one-dimensional correlations was revealed in these materials - in spite of their nominally two-dimensional triangular-lattice structures. On the other hand, in the manuscript we show the temperature and angular dependences of ESR fields in CROC. More systematic investigations of temperature and angular dependences of ESR parameters (including high-temperature regime, $T \gg J/k_B$, where the linewidth along different directions is dominantly determined by DMI) can be indeed an important topic of further experimental and theoretical investigations. We mentioned that in the manuscript and we do plan these experiments.

(2) Magnetization: The numerical simulation based on the determined magnetic parameters should be compared with the experimental data.

Our comment: We would like to thank Reviewer for this very useful remark. Following this suggestion, we have included results of recently published calculations employing the orthogonalized finite-temperature Lanczos method (OFTLM) for a triangular-lattice AF with parameters similar to CROC (Ref. 32). We obtained a very good agreement between the experimental data and the calculation results (Fig. 5), revealing the validity of the proposed model.

REVIEWERS' COMMENTS

Reviewer #1 (Remarks to the Author):

In the revised manuscript Zvyagin et al. have convincingly addressed the issues raised in the previous review round. We'd like to thank the authors for clarifying their measurements and their description as well as for carrying out additional measurements. We especially appreciate additional magnetization measurements with several different techniques, that nicely overlap with each other and with the destructive coil data, eliminating the magnetization curve uncertainties. Particularly useful is the addition of Figure 7 which supports the authors' analysis that DM vector is pointing along the c-axis of the material. Also important is the new magnetization curve, Fig.5, which contains an intriguing hint of a magnetization plateau in the 40-50 T interval. We hope the authors will continue their experimental investigation of this interesting possibility in the future.

We find the revised manuscript to be clear and well written and to contain interesting new physics. The manuscript is essentially ready for publication in Nature Communications. We would only like to point out that the explanation of the DMI origin (end of the red block at page 2) contains a confusing formulation. The text says "...contains DMI in the b direction", while a better thing to say would be "...contains DMI on the b-bond". This would help to avoid the confusion between the bond geometric alignment and the direction of corresponding Dzyaloshinskii-Moriya vector.

Another small issue is that the newly added angular dependence in Fig. 7 does not seem to be referenced in the main text. Also it would be nice to see a short description of this figure, other than the caption, in the body of the "Supplemental Material".

To summarize, the study contains a lot of interesting and new data on a novel intricate material. The data has not only been collected through a series of challenging experiments, but also understood and interpreted convincingly. It significantly advances our understanding of particular J-J' model, and hence advances the field of quantum magnetism in general. Thus, we would advise publication of the manuscript in Nature Communications with the small revisions mentioned above.

Reviewer #2 (Remarks to the Author):

In the revised manuscript Zvyagin et al. have convincingly addressed the issues raised in the previous review round. We'd like to thank the authors for clarifying their measurements and their description as well as for carrying out additional measurements. We especially appreciate additional magnetization measurements with several different techniques, that nicely overlap with each other and with the destructive coil data, eliminating the magnetization curve uncertainties. Particularly useful is the addition of Figure 7 which supports the authors' analysis that DM vector is pointing along the c-axis of the material. Also important is the new magnetization curve, Fig.5, which contains an intriguing hint of a magnetization plateau in the 40-50 T interval. We hope the authors will continue their experimental investigation of this interesting possibility in the future.

We find the revised manuscript to be clear and well written and to contain interesting new physics. The manuscript is essentially ready for publication in Nature Communications. We would only like to point out that the explanation of the DMI origin (end of the red block at page 2) contains a confusing formulation. The text says "...contains DMI in the b direction", while a better thing to say would be "...contains DMI on the b-bond". This would help to avoid the confusion between the bond geometric alignment and the direction of corresponding Dzyaloshinskii-Moriya vector.

Another small issue is that the newly added angular dependence in Fig. 7 does not seem to be referenced in the main text. Also it would be nice to see a short description of this figure, other than the caption, in the body of the "Supplemental Material".

To summarize, the study contains a lot of interesting and new data on a novel intricate material. The data has not only been collected through a series of challenging experiments, but also understood and interpreted convincingly. It significantly advances our understanding of particular J-J' model, and hence advances the field of quantum magnetism in general. Thus, we would advise publication of the manuscript in Nature Communications with the small revisions mentioned above.

Reviewer #3 (Remarks to the Author):

The authors addressed properly my previous concerns. Now, I think the revised version of the manuscript is suitable for publication in Nature Communications.

02.10.2022

Dear Referees,

Thanks a lot for a good news regarding our manuscript "Dimensional reduction and incommensurate dynamic correlations in the $S=1/2$ triangular-lattice antiferromagnet $\text{Ca}_3\text{ReO}_5\text{Cl}_2$ ". Thank you very much also for the time spent with our manuscript.

Please find attached our revised manuscript. All the changes are made in accordance with the Referee's suggestions and are highlighted in the manuscript text in red:

Referees A/B: "We would only like to point out that the explanation of the DMI origin (end of the red block at page 2) contains a confusing formulation. The text says "...contains DMI in the b direction", while a better thing to say would be "...contains DMI on the b-bond". This would help to avoid the confusion between the bond geometric alignment and the direction of corresponding Dzyaloshinskii-Moriya vector."

Our response: We have change this text fragment to "As a result, there is no inversion center between neighboring Re atoms along the chains, allowing uniform DMI on the b -bond."

Referees A/B: "Another small issue is that the newly added angular dependence in Fig. 7 does not seem to be referenced in the main text. Also it would be nice to see a short description of this figure, other than the caption, in the body of the "Supplemental Material"."

Our response: We have added some discussion on the angular dependence of ESR excitations in CROC in Supplementary Information, including a short description of the Supplementary Fig. 1 We have twice referred Supplementary Fig. 1 in the main text of the manuscript.

We hope that now, after the revision, our work will be accepted for publication.

Please do not hesitate to contact me in case of any questions or further inquiries.

With best regards,

Sergei Zvyagin.